# Emulating Hyperspectral and Narrow-Band Imaging for Deep-Learning-Driven Gastrointestinal Disorder Detection in Wireless Capsule Endoscopy

**DOI:** 10.3390/bioengineering12090953

**Published:** 2025-09-04

**Authors:** Chu-Kuang Chou, Kun-Hua Lee, Riya Karmakar, Arvind Mukundan, Pratham Chandraskhar Gade, Devansh Gupta, Chang-Chao Su, Tsung-Hsien Chen, Chou-Yuan Ko, Hsiang-Chen Wang

**Affiliations:** 1Division of Gastroenterology and Hepatology, Department of Internal Medicine, Ditmanson Medical Foundation Chia-Yi Christian Hospital, Chia-Yi 60002, Taiwan; vacinu@gmail.com (C.-K.C.); 06155@cych.org.tw (C.-C.S.); 2Obesity Center, Ditmanson Medical Foundation Chia-Yi Christian Hospital, Chia-Yi 60002, Taiwan; 3Department of Trauma, Changhua Christian Hospital, No. 135, Nanxiao St., Changhua City 50006, Taiwan; 88847@cch.org.tw; 4Department of Mechanical Engineering, National Chung Cheng University, No. 168, University Rd., Min Hsiung, Chia-Yi 62102, Taiwan; karmakarriya345@gmail.com (R.K.); arvindmukund96@gmail.com (A.M.); 5Department of Biomedical Imaging, Chennai Institute of Technology, Sarathy Nagar, Chennai 600069, Tamil Nadu, India; 6Information Technology Department, Sanjivani College of Engineering, Kopargaon 423603, Maharashtra, India; prathamgade2324_it@sanjivanicoe.org.in; 7Computer Science and Engineering Department, Thapar Institute of Engineering & Technology, Patiala 147001, Punjab, India; dgupta1_be21@thapar.edu; 8Department of Internal Medicine, Ditmanson Medical Foundation Chia-Yi Christian Hospital, Chia-Yi 60002, Taiwan; cych13794@gmail.com; 9Department of Gastroenterology, Kaohsiung Armed Forces General Hospital, No. 2, Zhongzheng 1st. Rd., Lingya District, Kaohsiung City 80284, Taiwan; 10Department of Technology Development, Hitspectra Intelligent Technology Co., Ltd., Kaohsiung City 80661, Taiwan

**Keywords:** spectrum aided visual enhancer, wireless capsule endoscopy, gastrointestinal diseases, hyperspectral imaging, white light imaging, narrow band imaging, polyps, esophagitis, ulcerative colitis

## Abstract

Diagnosing gastrointestinal disorders (GIDs) remains a significant challenge, particularly when relying on wireless capsule endoscopy (WCE), which lacks advanced imaging enhancements like Narrow Band Imaging (NBI). To address this, we propose a novel framework, the Spectrum-Aided Vision Enhancer (SAVE), especially designed to transform standard white light (WLI) endoscopic images into spectrally enriched representations that emulate both hyperspectral imaging (HSI) and NBI formats. By leveraging color calibration through the Macbeth Color Checker, gamma correction, CIE 1931 XYZ transformation, and principal component analysis (PCA), SAVE reconstructs detailed spectral information from conventional RGB inputs. Performance was evaluated using the Kvasir-v2 dataset, which includes 6490 annotated images spanning eight GI-related categories. Deep learning models like Inception-Net V3, MobileNetV2, MobileNetV3, and AlexNet were trained on both original WLI- and SAVE-enhanced images. Among these, MobileNetV2 achieved an F1-score of 96% for polyp classification using SAVE, and AlexNet saw a notable increase in average accuracy to 84% when applied to enhanced images. Image quality assessment showed high structural similarity (SSIM scores of 93.99% for Olympus endoscopy and 90.68% for WCE), confirming the fidelity of the spectral transformations. Overall, the SAVE framework offers a practical, software-based enhancement strategy that significantly improves diagnostic accuracy in GI imaging, with strong implications for low-cost, non-invasive diagnostics using capsule endoscopy systems.

## 1. Introduction

GID comprises a broad spectrum of pathogenic abnormalities influencing the digestive tract, each with unique morphological, vascular, and inflammatory traits [1,2]. If these disorders go undetected in their early stages, they can lead to long-term impairment or become life-threatening cancer [3,4]. Early identification is thus not only advantageous but also essential for efficient clinical treatment [5,6,7]. Eight distinct categories of GID were considered in this analysis because of their high clinical relevance. Polyps, which arise from the mucosal surface as either pedunculated or sessile growths, are a common and clinically significant target for early detection due to their established role as precursors to colorectal cancer [8,9,10]. While many are benign, certain types of particularly adenomatous polyps carry a substantial risk of malignant transformation. Their endoscopic look can vary greatly in size, color, and surface texture; hence, efficient cancer prevention plans depend on their accurate detection [11]. A more complex class, dyed-lifted polyps, are treated during endoscopic mucosal resections (EMRs) using staining agents like indigo carmine or methylene blue to raise and improve the polyp from surrounding tissue [12,13,14]. However, this method changes the image’s color profile, which makes automated classification tasks more difficult and calls for strong image enhancement algorithms. Dyed-resection margins constitute another clinically important category [15]. Dyes are used to mark the resected area’s edge following polyp removal to guarantee no abnormal tissue remains free. It is difficult to differentiate these margins from normal or inflammatory mucosa using traditional white light imaging (WLI) because they often look like uneven, low-contrast areas. Esophagitis, an additional class featured in this dataset, is an inflammation of the esophagitis lining that may result from acid reflux, infections, or allergic reactions [16,17,18,19]. It is marked by mucosal erythema, friability, linear erosions, and, in advanced cases, ulcerations. Timely detection is essential, as untreated esophagitis can progress to more severe conditions, including Barrett’s esophagitis and esophageal adenocarcinoma [20,21]. Ulcerative colitis is a rare, chronic inflammatory disease of the colon and rectum, characterized by persistent mucosal ulceration, crypt distortion, bleeding, and loss of vascular pattern [22,23]. Its endoscopic features often overlap with those of other colitises and early neoplasia, making accurate differentiation challenging. The inflammation presents in a widespread, uneven pattern, often obscured by scarring, necessitating high-resolution imaging with strong contrast to ensure diagnostic accuracy. Normal cecum, Z-line, and pylorus are the other three classes; they are more important for contextual categorization and GI tract localization than diseases themselves, but they do represent anatomical features [24]. Usually used as a landmark for a full colonoscopy, the cecum is the first segment of the large intestine. It typically appears as a pale, glistening mucosa with visible anatomical landmarks such as the appendiceal orifice and ileocecal valve. The Z-line, or squamocolumnar junction, marks the transition between the esophageal squamous epithelium and gastric columnar epithelium. Pylorus, the muscular gateway between the stomach and duodenum, exhibits variable appearance depending on its state of contraction and must be carefully distinguished from ulcerations, neoplasms, or hypertrophic abnormalities. Including these normal classes in the dataset guarantees a more balanced model training and strengthens the pathological classification by avoiding false positives.

Early identification of these GIDs remains a great difficulty, with utmost clinical importance [25]. Under conventional WLI, most early-stage lesions cause only minor changes in mucosal structure, texture, and vascular patterns. Moreover, the way endoscopic images are interpreted depends much on the experience and knowledge of the clinician; thus, they remain quite subjective. Particularly for flat lesions, non-polypoid development, and inflammatory changes, the interobserver variability is high and diagnosis results often suffer. The emergence of wireless capsule endoscopy (WCE) as a revolutionary diagnostic alternative enables non-invasive, full-length imaging of the gastrointestinal tract, particularly the small intestine [26,27]. The procedure entails ingesting a capsule-sized device equipped with a miniature camera, which autonomously captures and transmits images as it traverses the GI tract. This method eliminates the need for sedation, intubation, or direct manipulation with an endoscope, making it particularly suitable for paediatric patients, elderly individuals, and those with contraindications to conventional endoscopy [28,29]. It relies on low-resolution, standard RGB WLI and lacks integrated enhancement technologies such as Narrow Band Imaging (NBI) [30,31]. Additionally, WCE does not support biopsies or therapeutic interventions, restricting its utility to diagnostic observation alone. The capsule’s passive transit through the GI tract results in inconsistent frame capture, motion artifacts, and occasional gaps in mucosal visualization. These limitations, coupled with the narrow spectral range of RGB imaging, impair both clinical interpretation and the performance of machine learning models trained on WCE data.

The traditional WLI method in endoscopy is widely used due to its simplicity and accessibility, as it captures the appearance of tissue using the red, green, and blue (RGB) spectrum [32]. It lacks the diagnostic accuracy and spectral clarity needed to expose minute vascular or epithelial differences, which are necessary to differentiate between benign and neoplastic lesions. It naturally generates diluted data since it compresses complicated spectral information into only three broad channels, affecting image contrast and limited diagnostic detail. To address these limitations, NBI has been introduced as an optical enhancement technique that improves mucosal and vascular visualization by utilizing two narrow bands of wavelengths 415 nm and 540 nm, which match the absorption peaks of hemoglobin, which provide selective illumination on GID and make the detection more effective. It is not available in WCE, though, and is only applicable to some endoscopic systems, including Olympus. Furthermore, NBI only employs two wavelengths; hence, it does not offer complete spectral information and may not be able to differentiate tissues with comparable hemoglobin content. Hyperspectral imaging (HSI) is a more extensive method of spectral imaging that captures hundreds of adjacent spectral bands in the visible and near-infrared spectrum. Every pixel in an HSI image has a complete spectral signature that allows for thorough tissue characterization, depending on inherent reflectance characteristics. It offers significant advantages, including the potential for real-time optical biopsy by capturing detailed spectral signatures of tissues. However, several factors have hindered its clinical adoption in endoscopy; for example, machines are typically large, expensive, and not yet integrated into conventional or capsule-based endoscopic platforms. Moreover, the high-dimensional nature of hyperspectral data introduces considerable computational overhead and redundancy, necessitating dimensionality reduction techniques such as PCA [33,34]. The substantial data volume generated per frame also presents challenges for real-time processing, further limiting the feasibility of HSI in routine clinical workflows. Maintaining compatibility with the RGB image format used in standard and capsule endoscopy, this study presents SAVE, a new technique meant to convert WLI into HSI data and replicate NBI [35]. Starting with spectral reconstruction using Macbeth Color Checker calibration, the SAVE algorithm converts the RGB color space into CIE 1931 XYZ space by gamma correction and linearization. The most important spectral features are then obtained by means of PCA, lowering dimensionality while maintaining necessary diagnostic information. The CIEDE2000 metric is used for additional color correction to guarantee the simulated outputs correspond with actual Olympus NBI images. Without any hardware modification, this change helps standard and capsule endoscopy systems to approximate the diagnostic benefits of NBI and HSI. SAVE has the potential to greatly progress the field of gastrointestinal diagnostics and help to improve patient outcomes worldwide by means of enhanced picture contrast, improved classification accuracy, and early illness detection.

The suggested methodological innovation and validation framework in GI imaging is far more comprehensive than prior studies, which predominantly concentrated on polyp-specific datasets or one-class endoscopic classification. The integration of Macbeth Color Checker calibration, PCA-based spectral reconstruction, and regression error correction enabled us to guarantee that SAVE photos closely resemble Olympus NBI while maintaining high spectral fidelity. This study further extends the applicability of SAVE to a cross-architecture evaluation of MobileNetV2, MobileNetV3, Inception-NetV3, and AlexNet utilizing the Kvasir-v2 dataset, comprising 6490 photos across eight clinically distinct categories. The results indicate substantial improvements in classification accuracy across multiple categories, with MobileNetV2 achieving a superior F1-score of 96 in polyp detection, while AlexNet demonstrated an increase of up to 18 percent in F1-score classification for each lesion type. The translational benefit of the software-only SAVE technology is its ability to integrate into the existing workflow of wireless capsule endoscopy, which is presently constrained in NBI or HSI functionalities. The methodological advancements, reported experimental results, and translational applications collectively affirm that the current research significantly surpasses previous reports, presenting a feasible and scalable solution for enhancing non-invasive gastrointestinal diagnostics.

## 2. Materials and Methods

### 2.1. Dataset

This study works on the publicly available GI endoscopy image dataset “Kvasir-v2 dataset” created by Simula Research Laboratory in association with Vestre Viken Health Trust, Norway [36,37]. It has 6490 annotated images in eight classes: polyp, dyed-lifted polyp, dyed-resection margin, esophagitis, ulcerative colitis, normal cecum, pylorus, and Z-line. Medical experts have annotated and classified every image, which qualifies the dataset for supervised deep learning projects in GI image classification. Several preprocessing steps were implemented to prepare the dataset for model training and evaluation. Convolutional neural networks (CNNs) used input images that were resized to 224 × 224 pixels. To standardize image intensity across all inputs, pixel values were brought into line with the [0, 1] range. Data augmentation methods were used to solve class imbalance and enhance models’ generalizing capability. These comprised random zoom, minor translations, horizontal and vertical flipping, and small rotation (±15°). Only the training set was augmented to prevent data leakage. To guarantee that every subset included a proportional representation of all eight classes, the dataset was stratified and split into training, validation, and test sets at a ratio of 70:15:15, helping to perform best on unseen data. Table 1 illustrates that all eight classes, encompassing normal anatomy and diverse pathologies, are adequately represented across the training, validation, and test divisions of the dataset. The stratified distribution of 6490 images guarantees that our model is trained and assessed on a balanced and comprehensive array of cases. Figure 1 presents a workflow diagram that clearly delineates each phase of our methodology, encompassing image preprocessing, dataset partitioning, SAVE spectral transformation, and qualitative comparison of WLI against virtual narrow-band outputs. This elucidates the process by which color checker calibration and PCA-based regression yield the transformation matrix employed for producing enhanced endoscopic images.

### 2.2. Spectrum Aided Vision Enhancer

Any color that is visible to the human eye can be depicted using RGB values. Different colors result from the numerous combinations of red, green, and blue components. Regarding HSI, the colors are determined by these principles in addition to the light absorption and reflection intensity. By means of a reflectance chart, the SAVE technique converts colors existing in an RGB image acquired by a digital camera into an HSI image. Consequently, the Macbeth Color Checker, also called X-Rite Classic, helps to calibrate the system. X-Rite Classic consists of six shades of grey, primary colors (red, green, and blue), secondary colors (cyan, magenta, and yellow), and 24-color patches [38]. To properly view the colors based on human vision, the images in the 24-color patch are transformed to the CIE 1931 XYZ color space that normalizes the RGB values to a narrower range and further linearizes them to the CIE 1931 color spectrum. The digital camera’s photos could be contaminated by some noise or error, so utilizing a variable matrix as stated in Equation (1) corrects the error. Equation (2) computed the fresh X, Y, and Z values following corrections. (1)$$\left[C\right]=\left[{XYZ}_{Spectrum}\right]\times pinv(\left[V\right])$$(2)$$\left[{XYZ}_{Correct}\right]=\left[C\right]\times [V]$$

The method converts spectrometer and camera colors into the XYZ color space. Equations (3)–(6), respectively, allow the spectrometer’s convergence of reflectance spectra into the XYZ color space to translate sRGB into the XYZ color spectrum on the camera side.(3)$$X=k\int_{400\mathrm{nm}}^{700\mathrm{nm}}S\left(\lambda \right)R\left(\lambda \right)\bar{x}\left(\lambda \right)d\lambda$$(4)$$Y=k\int_{400\mathrm{nm}}^{700\mathrm{nm}}S\left(\lambda \right)R\left(\lambda \right)\bar{y}\left(\lambda \right)d\lambda$$(5)$$Z=k\int_{400\mathrm{nm}}^{700\mathrm{nm}}S\left(\lambda \right)R\left(\lambda \right)\bar{z}\left(\lambda \right)d\lambda$$(6)$$k=100/\int_{400\mathrm{nm}}^{700\mathrm{nm}}S\left(\lambda \right)\bar{y}\left(\lambda \right)d\lambda$$

The dark current component of the imaging device is expressed by a set value. We derive the variable matrix V by standardizing the V_color_ and V_Non-Linear_ product together with V_Dark_. Standardizing is limited by up to the third order to prevent the situation of overcorrection. Along with a 24-color patch reflectance spectrum, Ocean Optics QE65000 (Dunedin, FL, USA) transforms color into the XYZ color system. The spectrometer first achieves XYZ values by first measuring the colors on a 24-color patch board. By means of a precise mathematical relationship established by means of the regression analysis procedure, color space conversion errors were minimized, so the transformation matrix (M) as optimized by considering sensor-specific variations in Equation (7). To account for sensor-specific deviations, a second regression analysis was performed, aligning the estimated XYZ values with reference spectrometer data (see Appendix A for the RMSEs of the XYZ values before and after calibration).(7)$$\left[M\right]=\left[Score\right]\times pinv(\left[V_{Color}\right])$$

Reflectance spectrum data (R_Spectrum_) then enables a development of a transformation matrix for colors. The correctness of the transformation between the reference XYZ values (from spectrometer data) and the estimated XYZ values (from camera RGB data) is measured by the score. R_Spectrum_ then undergoes PCA to identify six main components, sometimes referred to as PCs, which effectively justified 99.64% of the data. This work mostly depends on normal CIE 1931 transformation values for the transformation matrix from sRGB to XYZ. Specifically in our imaging system, an experimental calibration using a color checker under controlled lighting conditions helped us to confirm the correctness. This calibration improved the transformation matrix by factoring spectral response variations and camera sensor characteristics, therefore guaranteeing consistency in color space conversion for the enhanced SAVE picture analysis. A transformation matrix was generated and connected with the PCA components, so a very low RMSE value of 0.056 and a color difference of 0.75 showed that great color similarity emerged. This approach effectively turns RGB photos taken into HSI images. The average chromatic aberration decreases from 10.76 to 0.63 after calibration, therefore improving color accuracy (see Appendix A for the color difference before and after camera calibration). The results revealed that red displayed the greatest variation in longer wavelengths between 600 and 780 nm, therefore indicating a restriction of this research from the reflectance variations between major colors, including red, green, blue, yellow, cyan, and magenta. Black is the one with the lowest RMSE value of 0.015; the other 23 color blocks have RMSE values less than 0.1; hence, the average RMSE is 0.056, demonstrating great color reproduction accuracy. The mean color difference was discovered to be 0.75, showing the visual accuracy of colour reproduction, when the RMSE values were visually and quantitatively expressed. This method transforms RGB photos into NBIs suitable for use in Olympus cameras to detect GID by use of HSI conversion. This guarantees that the images produced by this technique and the NBI images taken with the Olympus endoscope barely differed (see Appendix A for the RMSEs between analog and measured spectra of each color block). This color calibration is performed using the same 24-color checker. Following the matching between the SAVE-generated and real NBI, three main elements influencing color difference are mostly considered: the color matching function, the light function, and the reflection spectrum. Most of the light is absorbed in the 450–540 nm range, so a notable variation in the intensity of wavelengths was observed there. Along with the annealing optimization mechanism provided by Equation (8), this light spectrum was calibrated using Cauchy–Lorentz visiting distribution (see Appendix A for the LAB values of the simulated and observed colors).(8)$$f\left(x;x_{0},\gamma \right)=\frac{1}{\pi \gamma \left[1+{\left(\frac{x-x_{0}}{\gamma }\right)}^{2}\right]}=\frac{1}{\pi }\left[\frac{\gamma }{{\left(x-x_{0}\right)}^{2}+\gamma^{2}}\right]$$

Classical simulated annealing (CSA) is simplified here into fast annealing (FSA). The color difference was thus limited to a meager value of 5.36. Though the maximal hemoglobin absorption level was observed at 415 nm and 540 nm, the Olympus endoscope also detected traces of brown shade matching the wavelength of 650 nm in the genuine NBI image. Thus, to improve GID detection, which accounted for minor post-processing effects, other wavelengths, including 600 nm, 700 nm, and 780 nm, were also incorporated into the calibration process. This improves the matching between the calibrated photos and actual NBI images. While entropy averaged 0.37%, SSIM for the SAVE pictures increased to 94.27%. With a PSNR of 27.88 dB, the Olympus images validated the accuracy of the spectral conversion method and its implementation in medical imaging. Figure 2 and Figure 3 provides a visual overview of the six target categories in our dataset, highlighting the distinctive endoscopic features that drive accurate class differentiation in the WLI and SAVE imaging modalities, respectively (see Appendix A for more detailed information about color tables and SAVE).

### 2.3. Model Architecture

#### 2.3.1. Inception-Net V3

Inception-Net V3 is a deep CNN that is widely used for image classification due to its high accuracy and its stability [39]. It scales up the convolution network without adding extra computational costs. This model extensively uses factorized and asymmetric convolutions in the inception modules, which replace larger convolutions with smaller ones such as 3 × 3 or 1 × 3 followed by 1 × 3, which reduces the computational cost. It includes dimension reduction through 1 × 1 before performing expensive operations, which optimizes both the speed and size of the model. As it is built on the original inception framework, it inherits features like multiple stacked Inception modules with nuanced modifications. The network starts with convolutional layers followed by several Inception modules arranged at various spatial resolutions. The final layers include average pooling, dropout, and a dense soft max classifier for 1000 classes (ImageNet). 

#### 2.3.2. MobileNetV2 

MobileNetV2 is a cutting-edge, lightweight CNN that has been specifically engineered to achieve high-performance image classification with minimal computational overhead [40,41]. While still attaining competitive accuracy compared to larger models, it is especially suited for deployment in settings with limited resources, such as mobile or embedded medical devices. For GI image classification, where real-time processing and efficiency are vital, especially in WCE, MobileNetV2 offers an ideal mix of speed and accuracy. It uses the ideology of inverted residual blocks and linear bottlenecks. MOBILETV2 uses inverted residuals, expanding the input features before applying the transformation and then projecting them back to a compact representation, unlike conventional residual connections, in which identity mappings are applied across high-dimensional feature spaces. Every block comprises a 1 × 1 pointwise convolution for expansion, a 3 × 3 depthwise convolution for spatial filtering, and a 1 × 1 linear projection for dimensionality reduction [42].

#### 2.3.3. MobileNetV3 

MobileNetV3 is the most recent and sophisticated variant in the Mobile Net series, specifically engineered to optimize the trade-offs between accuracy, latency, and model size [43]. This makes it particularly well-suited for real-time, edge-based medical applications such as WCE. Having a strong foundation of MobileNetV1 and V2, MobileNetV3 presents a set of architectural improvements guided by neural architecture search (NAS) and tuned for mobile and embedded environments. It blends modern architectural innovations, including squeeze and excitation (Se) modules, hard-swish activation, and automated block selection using NAS with the inverted residual blocks and linear bottlenecks of MobileNetV2. Every inverted residual block comprises a projection layer, a lightweight 3 × 3 depthwise convolution, a with attention module, and an expansion layer, crucial for the capture of subtle GID. MobileNetV3 also replaces the ReLU6 activation with hard swish, a computationally efficient approximation of the swish activation function. This improves the model’s non-linearity while maintaining speed, contributing to better accuracy, particularly in fine-grained classification tasks. 

#### 2.3.4. AlexNet 

The AlexNet architecture has been a groundbreaking CNN that has significantly contributed to the development of deep learning for computer vision [44,45,46]. Its simplicity, robustness, and effective training behavior on rather small datasets make it relevant in medical imaging applications, even if it is rather shallow compared to contemporary networks. Five convolutional layers and then three fully connected layers make up the architecture. Particularly crucial in separating normal from pathological tissue patterns in endoscopic images, the first convolutional layers learn low- to mid-level spatial features, including edges, textures, and contours. Layers of max pooling interweave to progressively shrink spatial dimensions while still preserving salient features. Applied across the network, Relu (Rectified Linear Unit) activations add non-linearity, accelerating convergence during training. AlexNet uses dropout regularization in the fully connected layers to reduce overfitting. 

## 3. Results

This study is based on a series of experiments performed using four deep learning architectures, Inception-Net V3, MobileNetv2, MobileNetv3, and AlexNet, in order to assess the performance of the proposed SAVE algorithm and its impact on GI image classification. Every model was tested and trained on the SAVE-enhanced dataset produced via spectral reconstruction, as well as the original WLI dataset. Particularly in identifying minor or visually complex GI conditions such polyps, ulcerative colitis and dyed-resection margins, the main goal was to evaluate whether the spectral augmentation introduced by SAVE could improve classification accuracy. Calculated both per class and over the entire dataset, performance was assessed using conventional classification metrics, including accuracy, precision, recall, and F1-score. Furthermore, image quality metrics, including SSIM, PSNR, and entropy, were used to evaluate the fidelity of the SAVE generated images against their WLI counterparts. The comparison study not only shows the diagnostic value added by the SAVE transformation but also emphasizes how different model architectures react to enhanced spectral input. The efficacy of SAVE in improving both machine interpretability and diagnostic performance across several GID classes is shown in the following sections by comprehensive quantitative and visual results (see Appendix A for more detailed information about the results of models).

### 3.1. MobileNetV2

MobileNetV2 exhibited robust classification capabilities in both SAVE-enhanced datasets and WLI datasets (see Appendix A for the confusion matrix of MobileNetV2 for the WLI image dataset). Table 2 shows MobileNetv2′s general classification accuracy of almost 90% on both imaging modalities, demonstrating its resilience and adaptability to many spectral inputs (see Appendix A for the loss and accuracy history of MobileNetV2 for the WLI image dataset). Especially for classes, most importantly polyps, where the F1-score rose to 96% using SAVE, the SAVE dataset produced noticeably better results than WLI (see Appendix A for the classification Report of MobileNetV2 for the WLI image dataset). This development emphasizes how well the model uses the improved spectral characteristics given by the SAVE technique, especially in cases of lesions with faint contrast in normal RGB imaging (see Appendix A for the confusion matrix of MobileNetV2 for the SAVE image dataset). SAVE also outperformed WLI in classifying ulcerative colitis and dyed-resection margins, both of which depend mostly on fine-grained texture and vascular detail enhanced in the simulated hyperspectral bands (see Appendix A for the loss and accuracy history of MobileNetV2 for the SAVE image dataset). The performance between WLI and SAVE remained similar, nevertheless, for anatomically stable or high-contrast structures, including the Z-line, pylorus, and normal cecum (see Appendix A for the Classification Report of MobileNetV2 for the SAVE image dataset).

### 3.2. MobileNetV3

Maintaining an overall accuracy of 88%, MobileNetV3 showed regularly good classification performance on both the WLI and SAVE enhanced datasets (see Appendix A for the Confusion Matrix of MobileNetV3 for the WLI image dataset). With notable strengths and some subtle variations between the two imaging modalities, the model displayed balanced precision and recall across most classes as shown in Table 3 (see Appendix A for the Loss and Accuracy History of MobileNetV3 for the WLI image dataset). It recorded especially high performance in the dyed lifted polyp and dyed resection margin classes under the WLI configuration, both with an F1-score of 92%, demonstrating the model’s capacity to faithfully capture textural and contrast-based features in standard RGB inputs (see Appendix A for the classification report of MobileNetV3 for the WLI image dataset). With a recall of 93% and an F1-score of 89%, polyp classification also performed admirably (see Appendix A for the confusion matrix of MobileNetV3 for the SAVE image dataset). While achieving F1-scores of 80% in WLI and 79% in SAVE the results generally show that MobileNetV3 is well suited for leveraging the spectral richness of SAVE while retaining high performance on standard WLI even if esophagitis remained a rather difficult class to classify in both setups (see Appendix A for the loss and accuracy of MobileNetV3 for the SAVE image dataset and Appendix A for the classification report of MobileNetV3 for the SAVE image dataset).

### 3.3. Inception-Net V3

Inception-Net V3 was assessed on the SAVE-enhanced images, as well as the original WLI dataset, to find out how spectral augmentation affected classification accuracy (see Appendix A for the confusion matrix of Inception-Net V3 for the WLI image dataset). Table 4 shows that, using Inception-Net V3, WLI images routinely produced better overall performance with an accuracy of almost 89% (see Appendix A for the loss and accuracy of Inception-Net V3 for the WLI image dataset). By contrast, using the SAVE dataset produced an accuracy of 86% (see Appendix A for the classification report of Inception-Net V3 for the WLI image dataset). But when the performance of each class was looked at separately, SAVE clearly performed better in some areas (see Appendix A for the confusion matrix of Inception-Net V3 for the SAVE image dataset). In particular, SAVE-enhanced images helped to better classify ulcerative colitis and polyps, implying that the extra spectral detail added by the SAVE technique enhanced the model’s capacity to identify features, sometimes subtle under standard WLI (see Appendix A for the loss and accuracy history of Inception-Net V3 for the SAVE image dataset). On the other hand, WLI-based classification outperformed SAVE for structurally significant classes, including esophagitis, dyed-lifted polyps, and Z-line, most likely because Inception-Net V3’s deeper architecture favored high-contrast spatial features over spectral variations (see Appendix A for the classification report of Inception-Net V3 for the SAVE image dataset). These findings show that although Inception-Net V3 is quite good with conventional imaging, spectral augmentation has more benefits for inflammation-related or texture-dependent conditions.

### 3.4. AlexNet

AlexNet demonstrated a significant enhancement in performance when applied to SAVE enhanced images in comparison to standard WLI (see Appendix A for the confusion matrix of AlexNet for the WLI image dataset). Table 5 shows that general accuracy rose from 81% with WLI to 84% with SAVE, so stressing the value of spectral augmentation even with a rather small architecture (see Appendix A for the loss and accuracy history of AlexNet for the WLI image dataset). It performed reasonably in the normal (merged) class (F1-score: 88%) and somewhat less on polyps and esophagitis under WLI (see Appendix A for the classification report of AlexNet for the WLI image dataset). With F1-scores of 73% and 71%, respectively, it battled more difficult classes, including dyed-lifted polyps and dyed-resection margins (see Appendix A for the confusion matrix of AlexNet for the SAVE image dataset). The model showed significant improvement in almost all aspects using SAVE (see Appendix A for the loss and accuracy of AlexNet for the SAVE image dataset). Reaching F1-scores of 90%, 91%, and 88%, respectively, polyp, dyed-lifted polyp, and dyed-resection margin classification clearly improved. Ulcerative colitis also sprang from 75% to 85% (see Appendix A for the classification report of AlexNet for the SAVE image dataset).

## 4. Discussion

This study illustrates the substantial influence of spectral image enhancement on the classification accuracy of GID. SAVE tackles important constraints of conventional endoscopic imaging, i.e., its limited spectral resolution and poor contrast in small mucosal or vascular abnormalities, by converting ordinary WLI into simulated HIS- and NBI-like outputs. When compared to their WLI equivalents, across all four deep learning models assessed, Inception-Net V3, MobileNetv2, MobileNetv3, and AlexNet SAVE, enhanced photos either matched or improved classification performance. Crucially, these changes were especially noticeable in demanding disease categories, including ulcerative colitis dyed-resection margins and dyed-lifted polyps. Usually lacking contrast and modest inflammatory signals in traditional imaging, these classes are challenging to find with conventional RGB-based feature extraction. Particularly about tissue texture and vascular variation, the spectral richness given by SAVE helped the models to more precisely learn class-distinctive patterns. With a remarkable 96% F1-score in polyp identification utilizing SAVE and high F1-scores in both the WLI and SAVE circumstances, MobileNetV2 was among the best-performing architectures. When backed by rich input data, this emphasizes how well the model balances architectural simplicity with great classification strength. Likewise, Mobile NetV3 made good and consistent performance across both imaging modalities, especially in ulcerative colitis and dyed-resection margins, where it made use of improved spatial and channel-wise characteristics supplied by SAVE. Whereas Inception-Net V3, a deeper and more sophisticated network, performed well on WLI with an overall accuracy of 88%, the same model showed no appreciable performance increase with SAVE overall. Using SAVE, however, improved classification in inflammatory diseases like esophagitis and ulcerative colitis on a per-class basis, suggesting that the method offered significant spectrum clues for classes when spatial structure alone may be inadequate. This implies that enhanced spectral input helps even high-capacity networks, especially in cases of discriminating against subtle lesions that are difficult to find under traditional imaging. Being the shallowest architecture, AlexNet showed the most advantage from SAVE. Its accuracy of 81% on WLI was somewhat poor, but with SAVE, it rose to 84%. Emphasizing SAVE’s function in increasing feature richness for simpler models, classes including dyed-lifted polyps and dyed-resection margins witnessed F1-score gains of up to 18%. In clinical settings involving edge or portable devices, where lightweight models like AlexNet are more practical because of their reduced computing demand, this is especially pertinent.

Clinically speaking, these outcomes are quite exciting. Usually limited to proprietary hardware, SAVE allows the modeling of enhanced imaging modalities through a software-based transformation of RGB inputs. This has important consequences for scenarios like WCE, where sophisticated imaging technologies like NBI or hyperspectral endoscopy are lacking. Clinicians can increase diagnosis sensitivity by including SAVE into the image capture or post-processing pipeline without changing current gear or sacrificing workflow effectiveness. Moreover, the use of SAVE in machine learning pipelines not only raises model accuracy but also guarantees dependability in GID classification with overlapping information. Conditions like ulcerative colitis and esophagitis, which could seem visually similar in WLI, become more distinctive with SAVE-enhanced inputs, for instance. This helps to improve clinical judgement and lower false positives. The general efficacy of SAVE in enhancing classification performance across architectures and GI classes indicates that spectral improvement should be given great thought for incorporation into endoscopic image analysis systems. SAVE offers a useful and significant development in the field of computer-aided GI diagnosis, given its interoperability with conventional RGB imaging systems, low processing overhead, and proven influence on diagnostic accuracy.

## 5. Conclusions

This study introduces SAVE as a clinically viable and robust solution for enhancing GI image analysis through simulated HSI and NBI equivalent. SAVE greatly improves visual contrast and feature depth by converting standard WLI into spectrally enriched representations, especially in cases when subtle inflammatory or structural changes are challenging to find. For important GI conditions, including polyps, ulcerative colitis, and dyed-resection margins, evaluations across four deep learning architectures, Inception-Net V3, MobileNetv2, MobileNetv3, and AlexNet, showcased consistent improvements in classification performance. Lightweight models like AlexNet, which most benefited from SAVE’s spectral augmentation, showed the clearest performance improvements. SAVE-enhanced datasets repeatedly outperformed WLI, improving F1-scores in challenging to classify categories and closing the performance difference between high- and low-capacity models. These results strongly support SAVE’s inclusion into practical diagnostic procedures, particularly in settings where hardware restrictions limit the use of built-in enhancement technologies such as NBI or HSI endoscopy. SAVE has the potential to increase present diagnostic capabilities without requiring extra hardware given its compatibility with current RGB imaging systems, including WCE. Adoption of it in clinical practice could result in early, more accurate diagnosis of GID, improving patient outcomes.

## Figures and Tables

**Figure 1 bioengineering-12-00953-f001:**
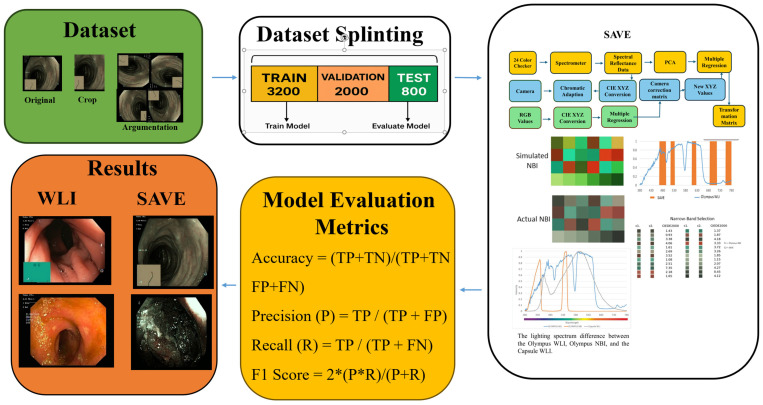
The study pipeline involves cropping and augmenting raw endoscopic frames, which are subsequently divided into training, validation, and test sets. Concurrently, a 24-patch color checker is captured using both a spectrometer and the endoscope. Its spectral reflectance is transformed through PCA and multiple regression into a camera-correction matrix, which converts white-light input into SAVE virtual narrow-band images. Representative outputs of WLI and SAVE are displayed side by side, and classification performance is evaluated using accuracy, precision, recall, and F_1_ score.

**Figure 2 bioengineering-12-00953-f002:**
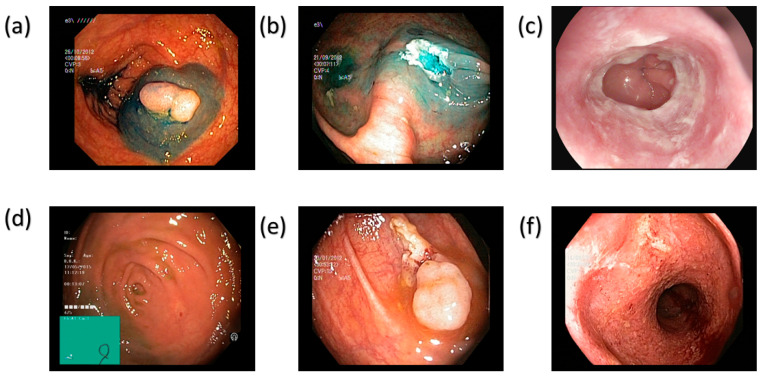
Representative endoscopic images for each of the six classes in the WLI modalities. (**a**) Dyed-lifted polyps—A polyp lifted by submucosal injection and stained with indigo-carmine dye. (**b**) Dyed-resection margins—The mucosal resection margin highlighted by dye after endoscopic removal. (**c**) Esophagitis—Erythematous, edematous mucosa consistent with inflammatory change in the esophagus. (**d**) Normal cecum—Unremarkable cecal mucosa with no pathological findings. (**e**) Polys—An untreated polyp under standard white-light endoscopy. (**f**) Ulcerative-colitis—Inflamed colonic mucosa with superficial ulceration and exudate.

**Figure 3 bioengineering-12-00953-f003:**
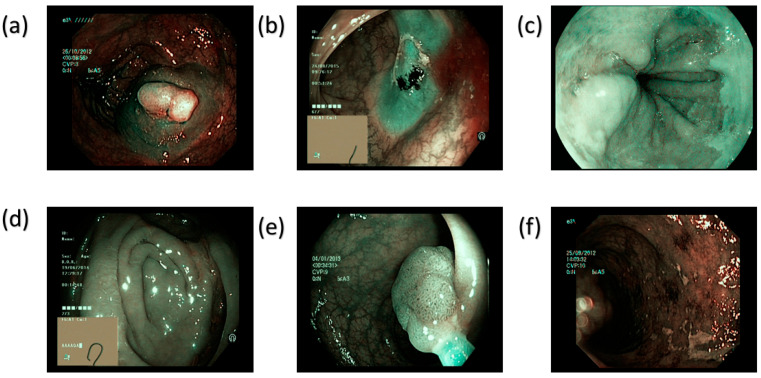
Representative endoscopic images for each of the six classes in the SAVE imaging modality after the conversion algorithm was applied. (**a**) Dyed-lifted polyps—A polyp lifted by submucosal injection and stained with indigo-carmine dye. (**b**) Dyed-resection margins—The mucosal resection margin highlighted by dye after endoscopic removal. (**c**) Esophagitis—Erythematous, edematous mucosa consistent with inflammatory change in the esophagus. (**d**) Normal cecum—Unremarkable cecal mucosa with no pathological findings. (**e**) Polys—An untreated polyp under standard white-light endoscopy. (**f**) Ulcerative-colitis—Inflamed colonic mucosa with superficial ulceration and exudate.

**Table 1 bioengineering-12-00953-t001:** Distribution of the 6490-image dataset among eight categories. For each category—dyed lifted polyps, normal z-line, dyed resection margins, normal pylorus, normal cecum, polyps, and ulcerative colitis—the distribution of images is as follows: training set (4535), validation set (1294), and test set (661), with per-class totals varying from 439 to 995 images.

Class Name	Train	Validation	Test	Total
Dyed Lifted Polyps	637	182	93	912
Normal Z-line	582	166	85	833
Dyed Resection Margins	366	104	55	525
Normal Pylorus	569	162	83	814
Normal Cecum	682	195	99	976
Polyps	696	199	100	995
Ulcerative Colitis	307	87	45	439
Esophagitis	696	199	101	996
Total	4535	1294	661	6490

**Table 2 bioengineering-12-00953-t002:** Classification Report of MobileNetV2.

Type	Classes	Precision	Recall	F1-Score	Accuracy
WLI	Normal (Merged)	91%	93%	92%	90%
	Polyps	88%	89%	89%	
	Dyed-Lifted Polyps	91%	90%	91%	
	Esophagitis	86%	82%	84%	
	Ulcerative Colitis	88%	93%	90%	
	Dyed-Resection Margins	94%	87%	91%	
SAVE	Normal (Merged)	91%	92%	92%	90%
	Polyps	94%	97%	96%	
	Dyed-Lifted Polyps	91%	90%	91%	
	Esophagitis	86%	82%	84%	
	Ulcerative Colitis	88%	93%	90%	
	Dyed-Resection Margins	94%	87%	91%	

**Table 3 bioengineering-12-00953-t003:** Classification Report of MobileNetV3.

Type	Classes	Precision	Recall	F1-Score	Accuracy
WLI	Normal	98%	100%	99%	92%
	Ulcer	98%	72%	83%	
	Polyps	80%	97%	87%	
	Esophagitis	98%	100%	99%	
SAVE	Normal	97%	100%	98%	95%
	Ulcer	89%	96%	93%	
	Polyps	97%	85%	91%	
	Esophagitis	99%	100%	99%	

**Table 4 bioengineering-12-00953-t004:** Classification matrix of Inception-Net V3.

Type	Classes	Precision	Recall	F1-Score	Accuracy
WLI	Dyed-Lifted Polyps	90%	77%	83%	89%
	Dyed-Resection Margins	77%	85%	81%	
	Esophagitis	77%	78%	78%	
	Normal (Merged)	91%	92%	92%	
	Polyps	81%	87%	84%	
	Ulcerative Colitis	85%	73%	79%	
SAVE	Dyed-Lifted Polyps	79%	81%	80%	86%
	Dyed-Resection Margins	76%	64%	69%	
	Esophagitis	78%	71%	75%	
	Normal (Merged)	90%	89%	89%	
	Polyps	84%	84%	84%	
	Ulcerative Colitis	83%	76%	79%	

**Table 5 bioengineering-12-00953-t005:** Classification report of AlexNet.

Type	Classes	Precision	Recall	F1-Score	Accuracy
WLI	Normal (Merged)	86%	90%	88%	81%
	Polyps	78%	80%	79%	
	Dyed-Lifted Polyps	79%	69%	73%	
	Esophagitis	77%	75%	76%	
	Ulcerative Colitis	79%	71%	75%	
	Dyed-Resection Margins	70%	73%	71%	
SAVE	Normal (Merged)	89%	90%	89%	84%
	Polyps	88%	92%	90%	
	Dyed-Lifted Polyps	91%	91%	91%	
	Esophagitis	80%	78%	79%	
	Ulcerative Colitis	88%	82%	85%	
	Dyed-Resection Margins	90%	85%	88%	

## Data Availability

The data presented in this study are available in this article upon considerable request to the corresponding author (H.-C.W.).

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
