# Peer review of "Emulating Hyperspectral and Narrow-Band Imaging for Deep-Learning-Driven Gastrointestinal Disorder Detection in Wireless Capsule Endoscopy"

_bioengineering, 2025, doi:10.3390/bioengineering12090953_

Round 1
Reviewer 1 Report
Comments and Suggestions for Authors
Although some parts of the article are highly technical and difficult to be followed by a doctor, the clinical implications are very clearly drawn by the authors, and the impact in increasing endoscopic diagnostic accuracy is high. This novel technique will be able to help WCE assessment in the future. Therefore I support the publication of this valuable research.
Reviewer 2 Report
Comments and Suggestions for Authors
This manuscript proposes a new framework SAVE for for deep learning–driven gastrointestinal disorder detection in WCE. The performance looks good, but there are many defects in it.
1.In Fig. 1, "Accuracy" equals "(TP+TN)/(TP+TN+FP+FN)", not "(TP+TN)(TP+TN+FP+FN)".
2.In Eq. (1), what are "C", "XYZ_spectrum", and "V"? What are their units?
3.In Eq. (1), what do the functions "[ ]" and "pinv" mean?
4.In Eq. (1), please show the sizes (columns and rows) of "[C]", "[XYZ_spectrum]", and "[V]" if "[ ]" denotes a matrix.
5.In Eq. (2), what is "XYZ_correct"? What is its unit?
6.What is the reference for Eqs. (1)-(2)?
7.In Eqs. (3)-(5), what do the functions "S", "R", "-x", "-y", and "-z" mean?
8.In Eqs. (3)-(5), the range for visible lights is 400-700nm. But, the range of real visible lights is 380-760nm. Is the integration range 400-700nm appropriate?
9.In lines 211-212, what are "V_color", "V_non-linear", and "V_dark"?
10.In Eq. (7), what are "M" and "Score"?
11.What is the reference for Eq. (7)?
12.In Eq. (8), what are "x", "x_0", and "gamma"? What are their units?
13.What is the reference for Eq. (8)?
14.What is the reference for "ResNet 50" in Sec. 2.3.1?
15.What is the reference for "MobileNetV2" in Sec. 2.3.2?
16.What is the reference for "MobileNetV3" in Sec. 2.3.3?
17.What is the reference for "Alex Net" in Sec. 2.3.4?
18.In Table 3, "WLI%" should be replaced by "WLI".
19.There are 8 classes of images, but only 5 classes of images are used in Tables 2-5. Why are the 3 classes "Normal Z-line", "Normal Cecum", and "Oesophagitis" not used?
In conclusion, there are almost no explanations for the parameters and functions of Eqs. (1)-(8). Major revision is necessary.
Reviewer 3 Report
Comments and Suggestions for Authors
The content of the manuscript shows striking overlap with an already published article:
https://pubmed.ncbi.nlm.nih.gov/40647664/
This article from the same last author is not cited. This is clearly a form of unacceptable self-plagiarism. Therefore, I recommend rejecting the publication.
Comments on the Quality of English LanguageDoes not apply
Round 2
Reviewer 2 Report
Comments and Suggestions for Authors
This revised manuscript is ready to be published.
Reviewer 3 Report
Comments and Suggestions for Authors
Dear authors,
Thanks for submitting a new version of the manuscript. However, even the latest version does not justify a publication. Substantially, there is not enough new data for a publication. This is a follow-up study with no new results and does not advance the field.
Best regards
